# A Traditional Korean Diet Alters the Expression of Circulating MicroRNAs Linked to Diabetes Mellitus in a Pilot Trial

**DOI:** 10.3390/nu12092558

**Published:** 2020-08-24

**Authors:** Phil-Kyung Shin, Myung Sunny Kim, Seon-Joo Park, Dae Young Kwon, Min Jung Kim, Hye Jeong Yang, Soon-Hee Kim, KyongChol Kim, Sukyung Chun, Hae-Jeung Lee, Sang-Woon Choi

**Affiliations:** 1CHA Bio Complex, CHA University, Seongnam 13488, Korea; pkshin34@gmail.com (P.-K.S.); rose6919@gmail.com (S.C.); 2Research Group of Healthcare, Korea Food Research Institute, Wanju 55365, Korea; truka@kfri.re.kr (M.S.K.); dykwon@kfri.re.kr (D.Y.K.); kmj@kfri.re.kr (M.J.K.); yhj@kfri.re.kr (H.J.Y.); shkim@kfri.re.kr (S.-H.K.); 3Department of Food Biotechnology, Korea University of Science and Technology, Daejeon 34113, Korea; 4Department of Food and Nutrition, College of BioNano Technology, Gachon University, Seongnam 13120, Korea; chris0825@hanmail.net; 5Department of Healthy Aging, GangNam Major Hospital, Seoul 06279, Korea; joyks71@gmail.com; 6Chaum Life Center, CHA University, Seoul 06062, Korea; 7Department of Nutrition, School of Public Health and Health Sciences, University of Massachusetts, Amherst, MA 01003, USA

**Keywords:** Korean diet, plasma microRNA, salivary microRNA, diabetes mellitus

## Abstract

The traditional Korean diet (K-diet) is considered to be healthy and circulating microRNAs (miRs) have been proposed as useful markers or targets in diet therapy. We, therefore, investigated the metabolic influence of the K-diet by evaluating the expression of plasma and salivary miRs. Ten women aged 50 to 60 years were divided into either a K-diet or control diet (a Westernized Korean diet) group. Subjects were housed in a metabolic unit-like condition during the two-week dietary intervention. Blood and saliva samples were collected before and after the intervention, and changes in circulating miRs were screened by an miR array and validated by individual RT-qPCRs. In the K-diet group, eight plasma miRs were down-regulated by array (*p* < 0.05), out of which two miRs linked to diabetes mellitus, hsa-miR26a-5p and hsa-miR126-3p, were validated (*p* < 0.05). Among five down-regulated salivary miRs, hsa-miR-92-3p and hsa-miR-122a-5p were validated, which are associated with diabetes mellitus, acute coronary syndrome and non-alcoholic fatty liver disease. In the control diet group, validated were down-regulated plasma hsa-miR-25-3p and salivary hsa-miR-31-5p, which are associated with diabetes mellitus, adipogenesis and obesity. The K-diet may influence the metabolic conditions associated with diabetes mellitus, as evidenced by changes in circulating miRs, putative biomarkers for K-diet.

## 1. Introduction

The traditional Korean diet (K-diet), which is rich in vegetables and fibers, moderate to high in legumes and fishes, and low in red meat [1], is regarded as a healthy diet in terms of metabolism [2,3]. A typical K-diet meal is served with a bowl of cooked white rice or mixed grains, a soup bowl with various types of broth and solid ingredients, and side dishes that are mostly seasoned with fermented soy products, herbs and sesame or perilla oil [1]. The most staple side dish of the K-diet is Kimchi, which is prepared from a variety of traditional fermented vegetables such as napa cabbage or daikon radish with salt, red pepper, garlic, and ginger. Ultimately, the K-diet is low in calories and fat, especially animal fat, but high in low glycemic index carbohydrates and vegetable protein. Due to its unique characteristics, the positive influence of the K-diet on metabolic disorders and cardiovascular disease risk has been widely investigated [4,5,6].

Previous cohort studies and clinical trials on the K-diet have been conducted using various anthropometric markers such as body weight and body mass index as well as blood markers such as total cholesterol, high density lipoprotein (HDL)-cholesterol, low density lipoprotein (LDL)-cholesterol, triglyceride, and fasting blood glucose [2,3,4,5,6]. However, studies investigating the molecular mechanisms by which the K-diet promotes positive health effects are rare, especially those based on microRNAs (miRs) as endpoints.

MiRs are a group of small, non-coding, single-stranded RNA molecules that are implicated in numerous human diseases due to their regulation of mRNA degradation or protein translation, ultimately affecting the expression of target genes [7]. Aberrant expression of miRs has also been detected in metabolic disorders such as obesity, insulin resistance, diabetes mellitus, and non-alcoholic fatty liver disease [8,9]. Due to their high stability, miRs released into the circulation can be utilized as the diagnostic or prognostic markers of certain health conditions [10,11]. A recent study suggested that circulating miRs can serve as predictive markers of the response of obese subjects to the low fat diet intervention [12]. Even though computational algorithms have been developed for the predictive power of miRs, it is still not enough because they are known to have high false-positive rates, and their predictions are not in agreement [13]. In this regard, we should be more cautious to use miRs as markers for diagnosis or prognosis.

As circulating miRs are much easier to collect than tissue miRs, they have been highlighted as non-invasive biomarkers present in body fluids [14]. Currently circulating miRs are important constituents of liquid biopsies, a new diagnostic concept that helps improve the monitoring and identification of tumor-associated molecules in body fluids [15].

Since miRs are also found in saliva, which can be collected non-invasively, interest in salivary miRs for the evaluation of certain diseases is increasing rapidly. A meta-analysis suggested that salivary miRs can represent a diagnostic marker for oral as well as systemic diseases [16]. The clinical and scientific evidence showing usefulness of saliva for evaluating systemic conditions or detecting systemic diseases will provide an easy and convenient diagnostic tool, especially for children and individuals with poor peripheral blood vessel conditions [17]. As most compounds present in blood can also be found in saliva, saliva is suggested to mirror the physiological state of the body [18]. In this regard, the use of saliva miRs has been extended to exercise evaluation [11] and forensic body fluid identification [19].

We, therefore, set up a hypothesis that the health effects of K-diet are conveyed through miRs, which can be detected in the circulation. To test the hypothesis, we analyzed the expression changes in circulating miRs using plasma and saliva samples collected from a two-week dietary intervention study [20] and determined the metabolic influence of the K-diet, through which K-diet may promote health conditions.

## 2. Materials and Methods

### 2.1. Subjects

Ten women aged 50 to 60 years were recruited and randomly and equally divided into either a K-diet group or control diet group (*n* = 5, each group) [20]. The major selection criteria were to have a body mass index (BMI) ranged between 25 and 30 kg/m^2^ and the blood total cholesterol between 200 to 270 mg/dL. The major exclusion criteria were individuals with diabetes mellitus, thyroid disorders, or cardiovascular diseases as well as individuals undergoing lipid lowering therapy or hormone replacement therapy. This human study was approved by CHA University Bundang Medical Center Institutional Review Board (BD 2015-107). Written informed consent was obtained from all subjects after receiving an explanation of the study.

### 2.2. Dietary Intervention

The participants in the K-diet group were provided the traditional K-diet, as reported in our previous study [1], while the control group was provided a Westernized Korean diet currently consumed in Korea. Both diets were prepared using traditional cooking methods. During the two weeks of dietary intervention, subjects were housed in a metabolic unit-like condition, where exercise and other activities were strictly controlled to minimize individual differences in energy expenditure, so that diet was the only metabolically distinct factor. Both diets followed the traditional Korean table setting with a rice bowl, soup, and side dishes. Both diet contained the same number of calories, but the K-diet consisted of more vegetables, vegetable protein and fat sources, and carbohydrates with a lower glycemic index, as well as less fat, especially animal fat, compared to the control diet. In addition, the K-diet group was provided with traditional drinks and soy milk, while the control diet group was provided with coffee, juice and milk. Dietary consumption was measured each day and consumed nutrients and calories were calculated using CAN-PRO 5.0 software (Korean Nutrition Society, Seoul, Korea). Alcohol consumption was not allowed but a small amount of alcohol was used for cooking.

### 2.3. Blood Chemistry

The blood chemistry profile of each subject was measured using an automated chemistry analyzer (Hitachi, Tokyo, Japan) at the Department of Clinical Pathology at Kangnam CHA hospital. Insulin resistance was determined using the homeostasis model assessment of insulin resistance (HOMA-IR), which was calculated using the equation: fasting insulin (mU/L) × fasting glucose (mg/dL)/405.

### 2.4. Dietary Glycemic Index (DGI) and Dietary Glycemic Load (DGL) 

DGI and DGL were calculated using a similar method as reported in the previous studies [21,22]. Among 285 foods used in this dietary intervention, 74 food items (26.0%) were mapped from the glycemic index (GI) database created for Korean foods [23]. GIs for seven food items (2.5%) were matched from the GI website of Sydney University (http://www.glycemicindex.com/) and GIs for 13 food items (4.6%) were imputed with similar food items. The remaining 191 (67.0%) food items that neither contained carbohydrate nor have a GI value were given a score of 0. DGI was calculated by summing the carbohydrate intake of each food multiplied by its GI value and then dividing by the total amount of carbohydrates consumed per day. DGL was calculated by multiplying the dietary GI value of each food item by the carbohydrate contents in each food, summing the product of each item, and then dividing by 100. Please refer to the following equations:(1)$$\mathrm{Dietary}\;\mathrm{glycemic}\;\mathrm{index}\;(\mathrm{DGI})\;=\;\;\sum_{i\;=\;1}^{n}\left(\mathrm{GI}\;\mathrm{value}\;\mathrm{of}\;\mathrm{food}\;\times \mathrm{available}\;\mathrm{carbohydrate}\;\mathrm{intake}\;\mathrm{from}\;\mathrm{food}\right)\;i\;/\;\mathrm{Total}\;\mathrm{available}\;\mathrm{carbohydrate}\;\mathrm{intake}\;\mathrm{in}\frac{g}{d}$$
(2)$$\mathrm{Dietary}\;\mathrm{glycemic}\;\mathrm{load}\;(\mathrm{DGL})\;=\;\;\sum_{i\;=\;1}^{n}\left(\mathrm{GI}\;\mathrm{value}\;\mathrm{of}\;\mathrm{food}\;\times \mathrm{available}\;\mathrm{carbohydrate}\;\mathrm{intake}\;\mathrm{from}\;\mathrm{food}\right)\;i\;/\;100$$

### 2.5. MiR Array for Screening Circulating miR

From the collected plasma and salivary samples, miRs were extracted using trizol and chloroform, and were purified using a spin column method (Qiagen, Hilden, Germany). The cDNAs were synthesized with the miScript II RT Kit (Qiagen, Hilden, Germany), using Takara Thermal Cycler Dice (Takara, Shiga, Japan), and utilized for the miR PCR array (Qiagen, Hilden, Germany), which quantifies 84 miRs that are most abundantly expressed in circulation using real time quantitative PCR (RT-qPCR) (Applied Biosystems, Carlsbad, CA, USA). The reaction mixtures were activated at 95 °C for 5 min, followed by 40 cycles of denaturation at 94 °C for 30 s, annealing at 57 °C for 30 s, and extension at 72 °C for 30 s. After amplification, the PCR products were confirmed through melting curve analysis. RNA U6 small nuclear gene (RNU6), which is the most commonly used internal control gene in miRNA RT-qPCR assays, was used as the reference gene, and the Ct values from each miR were normalized to the Ct value of RNU6.

### 2.6. Real Time Quantitative PCR (RT-qPCR) for Validation

To validate the screening results, individual RT-qPCRs were performed using cDNAs synthesized by the miScript II RT kit, miScript SYBR Green PCR kit and miScript Primer Assays kit (Qiagen, Hilden, USA). Briefly, 2× QuantiTect SYBR Green PCR master mix, 10× QuantiTect Primer assay containing target miRNA primer, and 10× miScript Universal Primer. RNase-free water and template cDNA were mixed. The expression of miRNAs was measured using RT-qPCR (Applied Biosystems, Carlsbad, CA, USA). The reaction mixtures were activated at 95 °C for 15 min, followed by 40 cycles of denaturation at 94 °C for 15 s, annealing at 55 °C for 30 s, and extension at 70 °C for 30 s. After amplification, the PCR products were confirmed through the analysis of the melting curve. The Ct values from each miR were normalized to the Ct value of RNU6. The miRs used for the validation are listed in Appendix A.

### 2.7. Statistics

Descriptive statistics are presented as means ± standard errors. The differences in miRNA expression between the two groups, before and after the intervention, were analyzed using a Student’s *t*-test or a paired *t*-test. Results with *p* < 0.05 were considered statistically significant. Data were analyzed using SPSS ver. 2.4 (IBM Corp., Armonk, NY, USA).

## 3. Results

### 3.1. Baseline Characteristics and Changes in Clinical Parameters

There was no significant difference in anthropometric markers nor blood chemistry parameters between the two groups before the dietary intervention (Table 1) (modified from [20]). After the two-week dietary intervention, the blood total cholesterol level was reduced in the K-diet group, while there was no change in the control group (Table 2) (modified from [20]).

### 3.2. Comparison of Macronutrient Intake and Food Consumption between the Two Diet Groups

Both groups consumed similar number of calories, but the K-diet group consumed more carbohydrates and proteins, especially plant based proteins, and less fat, especially animal fat, as well as less cholesterol, compared to the control group (Table 3) (modified from [20]). Moreover, the K-diet group consumed more whole grains, fruits and vegetables, legumes and tofu, fish and shells, nuts, and seaweed, while it consumed less red meat, eggs, and processed foods (Table 4) (modified from [20]).

### 3.3. DGI and DGL

The DGI of the K-diet was lower than that of the control diet (49.81 ± 0.24 vs. 54.35 ± 0.53 *p* < 0.0001). On the contrary, the DGL of the K-diet was higher than that of the control diet (149.98 ± 1.60 vs. 139.17 ± 2.82 *p* = 0.0012) (Table 3).

### 3.4. MiR Screening

Among 84 tested miRs, which are detectable in plasma, expression of 55 miRs was confirmed in plasma, of which eight were significantly down-regulated in the K-diet group, whereas one miR was up-regulated and the other was down-regulated in the control plasma (Table 5). In contrast, among the 84 tested miRs, 74 were expressed in saliva, of which five miRs were significantly down-regulated in the K-diet group, whereas two miRs were down-regulated and one was up-regulated in the control saliva (Table 6). Interestingly, plasma hsa-miR-148a-3p was up-regulated in the control diet, but down-regulated in the K-diet (Table 5). There was no overlap between plasma and saliva except with hsa-miR-25-3p, which was down-regulated in plasma and saliva samples of the control group.

### 3.5. Validation

Among eight down-regulated miRs screened in the plasma of the K-diet group (Table 5), hsa-miR-26a-5p and hsa-miR-126-3p were validated by RT-qPCR, whereas hsa-miR-25-3p, a miR down-regulated in control plasma, was validated (Figure 1). Among five down-regulated miRs screened in the saliva of the K-diet group (Table 6), hsa-miR-92-3p and hsa-miR-122-5p were validated, while hsa-miR-31-5p, another down-regulated in control saliva, was validated (Figure 2).

## 4. Discussion

Neither mutations nor polymorphisms can be reversed or changed by nutrition. However, nutrition can change gene expression through other molecular mechanisms without altering base pairs. Among the molecules involved in these mechanisms, miRs are one of the most abundant gene-regulatory molecules. MiRs have been extensively investigated to determine the molecular effects of nutrients, food and/or diet [37], and evaluating their potential may be useful to utilize them as biomarkers of health and diseases [38].

In recent years, increasing attention has been directed toward circulating miRs in blood and saliva, which may serve as useful diagnostic biomarkers, especially due to their remarkable stability and resistance to degradation. MiRs are easily transferred from organs to the circulation or from the circulation to organs, via microvesicles or exosomes, representing a novel cross talk model for cellular communication [39]. In the present study, eight plasma miRs were found to be down-regulated in the screening array of the K-diet group, and these were associated with type 1 and 2 diabetes mellitus, as well as prediabetes and gestational diabetes (Table 5). Some of these miRs were also associated with cholesterol metabolism and obesity, which might be the result of a decrease in total cholesterol level observed in the K-diet group. Among them, hsa-miR-26a-5p [24] and hsa-miR-126-3p [27] were validated by RT-qPCR (Figure 1), and our results show that these two miRs could serve as plasma markers in future K-diet studies. One validated miR in the control, hsa-miR-25-3p [24], was also associated with diabetes mellitus. Collectively, these observations suggest that the K-diet changes the expression of plasma miRs associated with diabetes mellitus, especially women at the age of 50 years without overt evidence of diabetes mellitus. Although it remains unclear how these miRs are involved in diabetes mellitus, future research is expected to focus on the relationship between the K-diet and diabetes mellitus-associated metabolism, using these miRs. The relationship with control diet will also add another layer of information.

Interestingly, plasma hsa-miR-148a-3p [24,25,26], which is associated with type 1 and type 2 diabetes mellitus, was up-regulated in the control diet group but down-regulated in the K-diet group. In a human study [40] that determined the association between the blood miR profile and the markers of glucose metabolism, hsa-miR-148a-3p was down-regulated in individuals with impaired fasting glucose and up-regulated in those with type 2 diabetes, compared to normal subjects. Further, hsa-miR-148a-3p has also been reported to be associated with the hemoglobin A1c level that reflects the average blood glucose level over approximately three months [40]. Even though this miR was not validated in either group, K-diet may have effects on diabetes mellitus associated metabolisms through this miR because it was the only miR that exhibited a polar response among the two diet groups. Thus, it is worthy of further evaluation.

Modulation of salivary miRs in response to K-diet was one of the novel observations made in the present study. Five salivary miRs that were down-regulated in the screening array were mostly associated with diabetes mellitus. Two down-regulated miRs and one up-regulated salivary miR in the control group were also associated with diabetes mellitus, obesity and adipogenesis (Table 6). Two down-regulated miRs, hsa-miR-92-3p [33] and hsa-miR-122a-5p [35], that were validated in the K-diet group, were associated with type 2 diabetes mellitus as well as non-alcoholic fatty liver disease and acute coronary syndrome. One down-regulated miR, hsa-miR-31-5p [32], validated in the control, was associated with adipogenesis and obesity. Since altered salivary miRs are linked to diabetes mellitus, it is not surprising that these miRs are also associated with diabetes mellitus-related metabolic conditions. The number of validated miRs was found to be less than that of screened miRs, possibly due to the limited subject numbers. If the study cohort is increased in future studies, a higher number of differentially regulated miRs may become clear.

Salivary glands are enveloped by capillaries and are highly permeable, allowing molecules to move into the saliva-producing cells and potentially influence the molecular constituents of the saliva [41]. Furthermore, saliva contains many different components including enzymes, antibodies, hormones, cytokines and microbiota that may provide additional information regarding health and disease status. In terms of liquid biopsy, saliva has many advantages over blood because collection is easy, safe and non-invasive. Salivary miR-based studies have tried to detect local and systemic diseases, including cancer [17]. Despite these favorable merits, the use of saliva as a diagnostic sample is yet to be a popularized, because blood serum or plasma is traditionally accepted and most frequently used source of measurable markers. Considering the advantage of using saliva as a source of biomarkers, including ease of sample collection due to its non-invasive nature, significant efforts must be undertaken to establish the efficacy of saliva-based markers to popularize its use as a standard diagnostic sample.

If the same miR changes were observed in plasma and saliva, it would have provided a more concrete evidence for the mechanism underlying the metabolic effects of the diet and a connection between the plasma miRs and saliva miRs. Unfortunately, we did not observe any overlap between plasma and saliva miRs, except for hsa-miR25-3p [24], which was validated only in plasma. In a previous study miR-25 was upregulated in type 1 diabetes patients and negatively associated with the residual beta cell function and positively associated with blood glucose control [24]. Thus, down-regulation of control diet might be associated with low glycemic load, even though our subjects were not diabetes mellitus patients. The control diet was not a Western diet but a Westernized Korean diet. As our IRB strongly suggested not providing an unhealthy diet as a control, we used a modest Western style control diet with less carbohydrate. Since the control diet has less carbohydrate and more fat, the down-regulation of this miR in the control group suggested that our control diet might have had influence on a certain segment of carbohydrate metabolism.

Even though there was no overlap between plasma and saliva in the K-diet group, most of the miRs were associated with diabetes mellitus, suggesting that miRs with similar functions were modulated by the K-diet. One plausible explanation for the lack of overlap of miR signature could be the differential expression level of miRs between plasma and saliva, even though salivary miRs are mainly derived from plasma. Collectively, the expression profile of salivary miR was somewhat different from that of plasma miR, and this difference may add one more layer of information that may not be covered by plasma miRs alone.

Since a single miR can regulate different genes and biological pathways, validated miRs may be involved in more diseases or disorders not evaluated in this study, such as cancer. Plasma hsa-miR-26a-5p has been associated with gastric cancer [42] and tissue hsa-miR-126-3p has been associated with non-small cell lung cancer [43]. Salivary miRs have not been commonly investigated in cancer research, but salivary hsa-miR-92-3p and hsa-miR-122a-5p have been reported as early detection markers for head and neck cancer [44]. Furthermore, circulating hsa-miR-92-3p [45] has been reported as a marker for tumor angiogenesis and endothelial mesenchymal transition differentiation. The altered miRs in the present study may be able to elucidate the link between cancer and metabolism, because metabolic disorders such as metabolic syndrome, obesity, diabetes mellitus, non-alcoholic fatty liver disease, and high fat diet are highly associated with the development of cancer [46]. One might speculate that altered miRs in the present study could be associated with the high incidence of cancer in Korea [47,48], but this hypothesis requires further evidence. As cancer is the leading cause of death in Korea [48], this would be an important area of future research.

Abnormal expression of oncogenes and tumor suppressor genes has been regarded as a major molecular mechanism underlying cancer development. MiRs can play a role in cancer development by affecting oncogenes and/or tumor suppressor genes, by either degrading targeted mRNAs or blocking translation [49]. MiR can also act as oncogenic miR by targeting tumor suppressor genes or as tumor suppressive miR by targeting oncogenes. On the contrary, miR expression can also be regulated by oncogenes and/or tumor suppressor genes, suggesting that a certain diet may retard or aggravate cancer development by regulating oncogenic and tumor suppressor miRs. This concept should be further explored in the future research based on the K-diet.

Circulating miRs that showed altered expression in response to the K-diet were associated with type 1 and 2 diabetes and prediabetes. In fact, the dietary glycemic index of the K-diet was lower than that of the control diet due to a higher intake of whole grains. Nevertheless, the dietary glycemic load was higher in the K-diet group due to a relatively high carbohydrate intake, which compensated for the low calorie intake from fat. In fact, the two diets were designed to have equal calorie intake. It appears that the expression of diabetes-associated miRs was altered to cope with the high carbohydrate load, which resulted in no significant differences in fasting blood glucose and insulin levels, as well as HOMA-IR (Table 2). Further, animal fat and cholesterol intake was lower in the K-diet group compared to the control diet group, which might have reduced the blood total cholesterol level in the K-diet group.

Even though this was a small pilot study, it provides credible evidence that the K-diet can significantly alter plasma and salivary miRs associated with diabetes mellitus and related metabolic conditions, along with reduced total cholesterol. These miRs could be useful for determining the metabolic effects of the K-diet in future studies and circulating miRs may serve as reliable biomarkers for the future diet-based studies.

## Figures and Tables

**Figure 1 nutrients-12-02558-f001:**
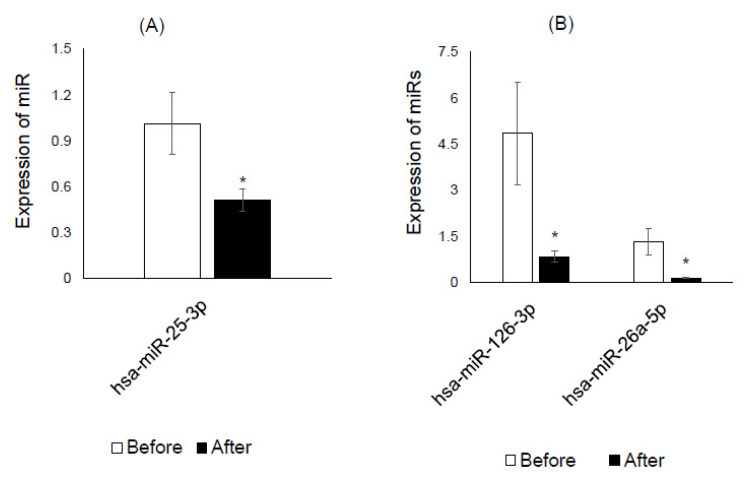
Validation of plasma miRs by RT-qPCR. (**A**) Hsa-miR-25-3p was down-regulated in the plasma of the control diet group. (**B**) Hsa-miR-126-3p and hsa-miR-26a-5p were down-regulated in the plasma of the K-diet group. White and black columns represent miR expression before and after the intervention, respectively. * *p* < 0.05 by paired *t*-test.

**Figure 2 nutrients-12-02558-f002:**
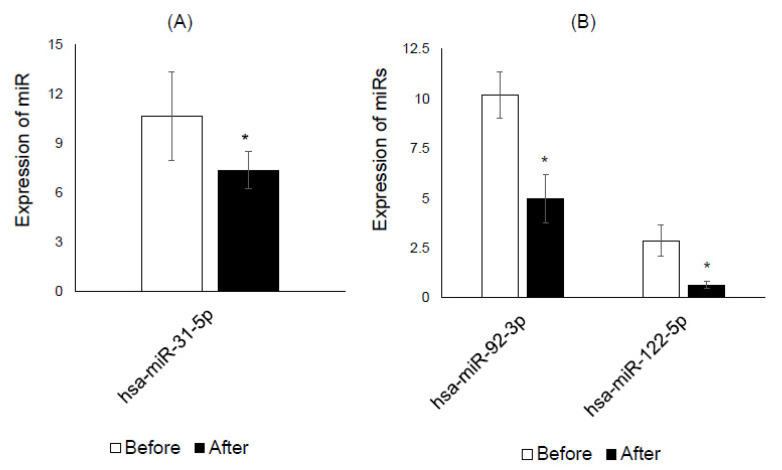
Validation of saliva miRs by RT-qPCR. (**A**) Hsa-miR-31-5p was down-regulated in the saliva of the control diet group. (**B**) Hsa-miR-92-3p and hsa-miR-122-5p were down-regulated in the saliva of the K-diet group. White and black columns represent miR expression before and after the intervention, respectively. * *p* < 0.05 by paired *t*-test.

**Table 1 nutrients-12-02558-t001:** Subject profile.

Variable	Control Diet (*n* = 5)	K-Diet (*n* = 5)	*p* Value
Age (years)	54.60 ± 0.87	52.80 ± 1.02	0.2165
Weight (kg)	66.48 ± 2.02	64.36 ± 2.12	0.4901
Waist circumference (cm)	90.60 ± 1.54	85.40 ± 3.64	0.2247
Systolic BP (mmHg)	130.80 ± 6.46	119.80 ± 4.00	0.1859
Diastolic BP (mmHg)	87.60 ± 4.08	78.20 ± 1.74	0.0670
Heart rate (bpm)	81.60 ± 2.84	74.80 ± 3.47	0.1673
Total Cholesterol (mg/dL)	209.40 ± 4.74	239.40 ± 15.14	0.1199
LDL-Cholesterol (mg/dL)	142.20 ± 5.40	138.40 ± 24.91	0.8881
HDL-Cholesterol (mg/dL)	46.56 ± 4.05	57.65 ± 3.40	0.3125
Triglyceride (mg/dL)	103.20 ± 12.99	237.80 ± 116.24	0.2165
Fasting glucose (mg/dL)	92.40 ± 4.15	93.40 ± 3.31	0.8554
Insulin (mU/L)	6.64 ± 1.05	10.46 ± 2.09	0.1415
HOMA-IR	1.48 ± 0.18	2.47 ± 0.55	0.143

BP: blood pressure, LDL: low density lipoprotein, HDL: high density lipoprotein, HOMA-IR: homeostasis model assessment of insulin resistance. All values are mean ± SE. The *p* value was determined by independent *t*-test.

**Table 2 nutrients-12-02558-t002:** Comparisons of clinical endpoints before and after the intervention.

Variable	Control Diet			K-Diet		
	Baseline	2-Week	*p* Value	Baseline	2-Week	*p* Value
Weight (kg)	66.48 ± 2.02	65.68 ± 1.97	0.9444	64.36 ± 2.12	63.36 ± 1.82	0.1481
Waist circumference (cm)	90.60 ± 1.54	88.20 ± 0.85	0.3224	85.40 ± 3.64	83.80 ± 2.05	0.5565
Total cholesterol (mg/dL)	209.40 ± 4.74	229.80 ± 8.12	0.1953	239.40 ± 15.14	198.20 ± 13.25	0.0163
LDL-cholesterol (mg/dL)	142.20 ± 5.40	146.60 ± 7.09	0.7568	138.40 ± 24.91	123.60 ± 13.05	0.5053
HDL-cholesterol (mg/dL)	46.56 ± 4.05	55.12 ± 5.53	0.0717	57.65 ± 3.40	49.52 ± 9.03	0.8180
Triglyceride (mg/dL)	103.20 ± 12.99	140.40 ± 16.10	0.1452	237.80 ± 116.24	125.40 ± 16.36	0.3600
Fasting glucose (mg/dL)	92.40 ± 4.15	89.60 ± 3.39	0.3719	93.40 ± 3.31	82.20 ± 3.92	0.1396
Insulin (mU/L)	6.64 ± 1.05	9.00 ± 1.20	0.5413	10.46 ± 2.09	7.40 ± 1.34	0.1023
HOMA-IR	1.48 ± 0.18	1.98 ± 0.24	0.129	2.47 ± 0.55	1.46 ± 0.20	0.143

LDL: low density lipoprotein, HDL: high density lipoprotein, HOMA_IR: homeostasis model assessment of insulin resistance. All values are mean ± SE. The *p* value was determined by paired *t*-test.

**Table 3 nutrients-12-02558-t003:** Comparison of macronutrient intake between the K-diet group and control.

Nutrients	Control Diet (*n* = 5)	K-Diet (*n* = 5)	*p* Value
Energy (kcal)	1775.5 ± 25.5	1740.2 ± 12.7	NS
Carbohydrate (% of energy)	57 ± 0.6	63.7 ± 0.4	<0.0001
Dietary glycemic index	54.35 ± 0.53	49.81 ± 0.24	<0.0001
Dietary glycemic load	139.17 ± 2.82	149.98 ± 1.60	0.0012
Protein (% of energy)	15.7 ± 0.2	17.1 ± 0.3	<0.0001
Animal based protein (% of energy)	7.3 ± 0.3	4.9 ± 0.3	<0.0001
Plant based protein	8.4 ± 0.1	12.2 ± 0.1	<0.0001
Fat (% of energy)	27.4 ± 0.4	19.2 ± 0.3	<0.0001
Animal based fat (% of energy)	10.4 ± 0.5	2.3 ± 0.2	<0.0001
Plant based fat (% of energy)	17.1 ± 0.3	16.9 ± 0.3	NS
Cholesterol (mg)	447.3 ± 30	182.9 ± 11	<0.0001

All values are mean ± SE. The p value was determined by independent *t*-test. NS: not significant.

**Table 4 nutrients-12-02558-t004:** Comparison of food consumption between the K-diet group and control (g/day).

Food Groups	Control Diet (*n* = 5)	K-Diet (*n* = 5)	*p* Value
Total grains	217.4 ± 5.1	277.7 ± 3.9	<0.0001
Whole grains	0.4 ± 0.1	267.9 ± 4.9	<0.0001
Vegetables and fruits	405.1 ± 7	543.2 ± 10.3	<0.0001
Kimchi	132.2 ± 4.3	160.9 ± 5.1	<0.0001
Legumes and tofu	40 ± 4.6	63.4 ± 4.6	0.0004
Nuts	2.6 ± 0.6	21.4 ± 3.8	<0.0001
Fishes and shell	35.4 ± 3.9	53.2 ± 5.3	0.0073
Seaweeds	15.7 ± 2.8	24.5 ± 3.9	0.0708
Meats	57.3 ± 4.6	10 ± 2.2	<0.0001
Red meats	48.8 ± 4.2	5.4 ± 1.3	<0.0001
Eggs	40.5 ± 4.1	7 ± 1.5	<0.0001
Processed foods	21.9 ± 3.4	0 ± 0	<0.0001
Salad dressing including mayonnaise	11.7 ± 0.6	0 ± 0	<0.0001

All values are mean ± SE. The *p* value was determined by independent *t*-test.

**Table 5 nutrients-12-02558-t005:** Plasma microRNAs (miRs) changed in the screening array.

Diet Group	miRNA	Expression Change	Associated Conditions	References
Control diet	hsa-miR-25-3p	Down	Type 1 diabetes	[24]
	hsa-miR-148a-3p	Up	Type 1 diabetes Type 2 diabetes	[24,25,26]
K-diet	hsa-miR-126-3p	Down	Prediabetes Type 2 diabetes	[27]
	hsa-miR-18a-5p	Down	Type 2 diabetes	[26]
	hsa-miR-19b-3p	Down	Gestational diabetes Cholesterol metabolism	[28,29]
	hsa-miR-107	Down	Type 2 diabetes Obesity	[30]
	hsa-miR-148a-3p	Down	Type 1 diabetes Type 2 diabetes	[24,25,26]
	hsa-miR-26b-5p	Down	Type 2 diabetes	[25]
	hsa-miR-374a-5p	Down	Type 2 diabetes	[31]
	hsa-miR-26a-5p	Down	Type 1 diabetes	[24]

hsa-miR: Homo sapiens-microRNA.

**Table 6 nutrients-12-02558-t006:** Saliva miRs changed in the screening array.

Diet Group	miRNA	Expression Change	Associated Disorders	References
Control diet	hsa-miR-25-3p	Down	Type 1 diabetes	[24]
	hsa-miR-31-5p	Down	Adipogenesis Obesity	[32]
	hsa-miR-200a-3p	Up	Type 1 diabetes	[24]
K-diet	hsa-miR-92-3p	Down	Type 2 diabetes Acute coronary syndrome	[33]
	hsa-miR-17-3p	Down	Type 2 diabetes Obesity	[34]
	hsa-miR-25b-3p	Down	Type 1 diabetes	[24]
	hsa-miR-122a-5p	Down	Type 2 diabetes NAFLD	[35]
	hsa-miR-193a-5p	Down	Type 2 diabetes	[36]

hsa-miR: Homo sapiens-microRNA, NAFLD: non-alcoholic fatty liver disease.

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
