# Peer review of "A Traditional Korean Diet Alters the Expression of Circulating MicroRNAs Linked to Diabetes Mellitus in a Pilot Trial"

_nutrients, 2020, doi:10.3390/nu12092558_

Round 1

Reviewer 1 Report

The authors investigated the influence of the traditional Korean diet (K-diet) on microRNAs that are related to diabetes mellitus. The study had 10 females divided in two groups K-diet and Westernized Korean diet.

Overall this is an interesting and well written paper. Below are a few points for improvements are

Introduction. More information about the predictive power of miRs on similar conditions should be added to the introduction. It is mentioned on lines 64, 65 but information about how well of a predictor should be provided.

Table 1. Although not significantly different, there is a lot of variability between the two groups at the beginning of the study. Example control diet avg TG= 103 mg/dL vs K-diet 237 mg/dL.  P-values should be provided for this analysis for each of the variables instead of NS.

There is a similar issue with table 2. P-values should be provided for each of the parameters.

Table 2. The variability of the markers from baseline to 2 weeks is not conventional TC average decreased almost 40 mg/dL. Is there evidence of similar drastic effects is such short term from previous studies?

Discussion. More specific information on what every specific influenced miRs influenced by the intervention does should be provided. “associated with type 1 diabete” is not sufficient. In which way it influence it, negatively? How?

Plasma hsa-miR-148a-3p. Provide an explanation of the role of this miR specially since it was the one showing a polar response.

Discussion. There is no clear explanation of the benefits observed between analyzing plasma vs saliva. What is the conclusion of this study comparing both methods, can any be drawn? Which one shown to be a better predictor of this chronic conditions.

Author Response

Response to Critiques from Reviewer 1

The authors investigated the influence of the traditional Korean diet (K-diet) on microRNAs that are related to diabetes mellitus. The study had 10 females divided in two groups K-diet and Westernized Korean diet.

Overall this is an interesting and well written paper. Below are a few points for improvements are

Critique 1) Introduction. More information about the predictive power of miRs on similar conditions should be added to the introduction. It is mentioned on lines 64, 65 but information about how well of a predictor should be provided.

Ans 1)

Thank you for the suggestion. The predictive power of miRs is very important.
The following sentence has been added to the line 66-69 with a reference.

“Even though computational algorithms have been developed for the predictive power of miRs, it is still not enough because they are known to have high false-positive rates, and their predictions are not in agreement (13). In this regard, we should be more cautious to use miRs as markers for diagnosis or prognosis.

Critique 2) Table 1. Although not significantly different, there is a lot of variability between the two groups at the beginning of the study. Example control diet avg TG= 103 mg/dL vs K-diet 237 mg/dL.  P-values should be provided for this analysis for each of the variables instead of NS.

Ans 2) p-values have been added to the table 1.

Critique 3) There is a similar issue with table 2. P-values should be provided for each of the parameters.

Ans 3) p values have been added to the table 2

Critique 4) Table 2. The variability of the markers from baseline to 2 weeks is not conventional TC average decreased almost 40 mg/dL. Is there evidence of similar drastic effects is such short term from previous studies?

Ans 4) In fact, there are very few K-diet intervention studies, which is one of the important reasons why this manuscript needs to be published. In a previous trial conducted by the USDA researchers (J Acad Nutr Diet. 2015 Jul;115(7):1083-92. doi: 10.1016/j.jand.2015.03.023), total cholesterol and LDL-cholesterol significantly decreased on the Korean diet (P<0.0001 and P<0.01, respectively) after 4 weeks. Further, in our crossover study with somewhat different objectives (not published yet), both total cholesterol and triglyceride was significantly decreased by a similar K-diet after 4 week intervention. Thus, we are convinced the data generated in this study. Please refer to Table 3 that demonstrated the lower consumption of animal fat and cholesterol in the K-diet group.

Critique 5) Discussion. More specific information on what every specific influenced miRs influenced by the intervention does should be provided. “associated with type 1 diabete” is not sufficient. In which way it influence it, negatively? How?

Ans 5) We are very careful to write such a concrete result. One miR can affect different genes, different pathways, and different diseases. At the same token, one gene can also be influenced by many different miRs. Thus, the expressional level alone does not seem to be sufficient to set up diagnosis or assess disease progression. The aim of the present study is to determine whether the K-diet can change the expression of circulating miRs or not, and what kind of health conditions those miRs are related to. Larger studies based on the result of this study may figure out their influence more specifically.

Critique 6) Plasma hsa-miR-148a-3p. Provide an explanation of the role of this miR specially since it was the one showing a polar response.

Ans 6) Yes, we paid a special attention to this miR but one concern we had is that this miR did not pass the validation. We rearranged the whole paragraph (Line 234-243).  

Critique 7) Discussion. There is no clear explanation of the benefits observed between analyzing plasma vs saliva. What is the conclusion of this study comparing both methods, can any be drawn? Which one shown to be a better predictor of this chronic conditions.

Ans 7) We also tried to determine which miRs is better, serum or saliva. Interestingly, serum expressed 55 miRs and saliva expressed 74 miRs among 84 tested miRs that are known to be detectable in plasma. On the other hand, the screening array found that 10 miRs in plasma and 8 miRs in saliva were significantly changed. Among them, 3 miRs in plasma and 3 miRs in saliva were validated and there was no overlap (Line 182-190, tables 5 and 6, and figures 1 and 2).  Based on this result we cannot determine which could be better but it seems that they may have synergistic effect for the prediction because the altered miRs were different between the two samples but their functions are similar.

One of specific aims for this experiment was to determine whether the K-diet can change the salivary and plasma miRs or not.  The next step will be determining whether these miRs are useful makers to determine the metabolic influence or finding the mechanism. 

Reviewer 2 Report

I would like to thank the authors of the manuscript for their contribution to the field of molecular mechanisms and diet.

The authors present interesting findings as it relates to miRNAs related to glucose metabolism and intake of the traditional Korean diet. The discussion of the results however are not framed in the context of the participant characteristics  - female, pre-menopausal, overweight, insulin resistant at baseline.  I also suggest that the authors include at least the gender of the participants in the title of the manuscript.

The authors also do not discuss if there was a washout period prior to the intervention diet or if the participant's pre-intervention usual diet were collected.

Author Response

Response to Critiques from Reviewer 2

I would like to thank the authors of the manuscript for their contribution to the field of molecular mechanisms and diet.

The authors present interesting findings as it relates to miRNAs related to glucose metabolism and intake of the traditional Korean diet.

Critique 1) The discussion of the results however are not framed in the context of the participant characteristics  - female, pre-menopausal, overweight, insulin resistant at baseline.

Ans 1) Thank you for the critique. As suggested it was added to the Line 230.

Critique 2) I also suggest that the authors include at least the gender of the participants in the title of the manuscript.

Ans 2) The gender information has been added to the title.

Critique 3) The authors also do not discuss if there was a washout period prior to the intervention diet or if the participant's pre-intervention usual diet were collected.

Ans 3) Unfortunately, there was no washout period. If we had such a washout period, it could have been great.

Reviewer 3 Report

The authors conducted a two week pilot study to determine the effects of the Korean diet (K -diet) on changes in circulating miRs in plasma and saliva. The findings showed eight plasma miRs were down-regulated of which two were linked to diabetes mellitus and five down-regulated salivary miRs were associated with diabetes mellitus, acute coronary syndrome and non-alcoholic fatty liver disease in the K - diet. In the control diet group, two down-regulated plasma and salivary miRs were validated and associated with diabetes and obesity. Total cholesterol was reduced by the K-diet. The authors concluded the K-diet may influence the metabolic conditions associated with diabetes mellitus, as evidenced by circulating miR changes. 

General comments:

The authors provide potentially interesting novel information regarding the impact of healthy diet on relationship between miRs and disease risk. However, it is abundantly clear the very small sample size in each group (n=5) does not provide sufficient power analysis to make any firm conclusions. It would behoove the investigators to recruit an additional 5 subjects in each group and present a more robust statistical analysis of their data. Lastly, the authors need to provide a stronger interpretation and rationale as to the perplexing favorable findings of the down-regulation of the hsa-miR-25-3p in the control group in both plasma and saliva. This appears to be completely contradictory to their findings.

Specific comments:

line 4  please include this was a pilot study in the title.

line 27  ten subjects seem much too low powered to detect signficance. Please consider adding n=5 to each group and re-analyzing the data.

line 36  were any of the down-regulated miR in control diet associated with disease risk? Please include in abstract as was done with K-diet.

line 80  please include a hypothesis based on previous literature.

line 89  is this registered as a clinical trial?

line 97  please correct, "bowl"

line 103  please include a table listing the ingredients of each diet compared to one another.

line 159  the sample size is small and control group started 20 mg/dL lower than k-diet which may have impacted the results. Please explain.

line 185  why was this miR down-regulated in control diet for both salivary and blood? Seems contradictory. Needs explaining.

line 192  both of these seem compelling for the control to lower risk for diabetes and obesity

line 206  This is not necessarily an accurate statement. Previous research suggests the possibility for food to alter dna sequence over time: Emily A. Seward, Steven Kelly. Dietary nitrogen alters codon bias and genome composition in parasitic microorganisms. Genome Biology, 2016; 17 (1) DOI: 10.1186/s13059-016-1087-9

line 214  do the authors have data on exosomes in the current study?

line 225  and control diet for that matter.

line 228  authors need to explain why it was not validated. seems critical to the basis of this entire paragraph.

line 244  authors need to explain why these were down-regulated in the control diet. seems very discordant with their findings.

line 246  authors support the need to increase the sample size prior to publication of these findings. 

line 263  how would this be direct evidence for the mechanism underlying the metabolic effects of the k diet if the down-regulation occurred in the control diet? the authors present a confusing statement as the seeming favorable result occurred in the control diet.

line 283  this entire paragraph seems overly speculative and not directly related to the objectives of this study and should be deleted.

line 291  this paragraph is also unnecessary and should be deleted.

line 299  This argument is not supported by their data. Indeed, the greater GL in the k diet would adversely affect insulin/glucose more than GI in most circumstances. The authors need a more thoughtful and appropriate explanation.

Author Response

Response to Critiques from Reviewer 3

The authors conducted a two week pilot study to determine the effects of the Korean diet (K -diet) on changes in circulating miRs in plasma and saliva. The findings showed eight plasma miRs were down-regulated of which two were linked to diabetes mellitus and five down-regulated salivary miRs were associated with diabetes mellitus, acute coronary syndrome and non-alcoholic fatty liver disease in the K - diet. In the control diet group, two down-regulated plasma and salivary miRs were validated and associated with diabetes and obesity. Total cholesterol was reduced by the K-diet. The authors concluded the K-diet may influence the metabolic conditions associated with diabetes mellitus, as evidenced by circulating miR changes.

General comments:

Critique 1) The authors provide potentially interesting novel information regarding the impact of healthy diet on relationship between miRs and disease risk. However, it is abundantly clear the very small sample size in each group (n=5) does not provide sufficient power analysis to make any firm conclusions. It would behoove the investigators to recruit an additional 5 subjects in each group and present a more robust statistical analysis of their data.

Ans 1) This is a pilot intervention study to determine whether K-diet can change circulating miRs in the plasma and saliva.  If we enroll 5 more subjects now to increase the statistical power, we cannot provide the exactly same diet and environment.

We rented a mansion building at the foot of the mountain for the intervention study. Although we have a small sample size, all subjects’ lifestyle and diet were controlled in detail through a 24-hour monitoring system every day, recorded by research managers who lived with them throughout the entire intervention period. In larger studies, it could be very difficult to control these things.

The number of researchers who resided with subjects and managed the experiment during the study period were 1 nurse, 1or 2 nutritionist, 2 cooks, and 2 study managers. The study was conducted in three laboratories and many researchers participated as coordinators and advocates. Please see our method section. All subjects were housed in a condition similar to the metabolic unit, where exercise and other activities were strictly controlled to make all conditions similar except diet. Diet and snack were prepared by dietitians. We weighed all foods and served and measured the leftover.

Since we already have found the fact that circulating miRs can be changed by K-diet, further validation can be done through a new study with more subjects and more specific focus on miRs related to diabetes mellitus and diabetes associated metabolism. Since we already demonstrated that salivary and plasma miRs can be changed by diet, anyone who is studying on diet can adopt this idea to their studies.

Critique 2) Lastly, the authors need to provide a stronger interpretation and rationale as to the perplexing favorable findings of the down-regulation of the hsa-miR-25-3p in the control group in both plasma and saliva. This appears to be completely contradictory to their findings.

Ans 2) Yes, the hsa-miR-25-3p is quite important in many health conditions. The whole paragraph has been rewritten and new sentences were added (Line 271 to 279).

“In a previous study miR-25 was upregulated in type 1 diabetes patients and negatively associated with the residual beta cell function and positively associated with blood glucose control [24], thus down-regulation of control diet might be associated with low glycemic load, even though our subjects were not diabetes mellitus patients. The control diet was not a Western diet but a Westernized Korean diet. As our IRB strongly suggested not providing an unhealthy diet as a control, we used a modest Western style control diet with less carbohydrate. Since the control diet has less carbohydrate and more fat, the down-regulation of this miR in the control group suggested that our control diet might have had influence on a certain segment of carbohydrate metabolism.”

Please refer to the following sentence from a review paper regarding miR 25 (Sárközy M, Kahán Z, Csont T. A myriad of roles of miR-25 in health and disease. Oncotarget. 2018;9(30):21580-21612).

“miR-25 acts as a double-edged sword in the development of diverse disease. Expressional level of miR-25 alone does not seem to be sufficient to set up diagnosis or assess disease progression. Expressional change of miR-25 among a specific set of miRNAs or other biomarkers might be more reliable in the diagnosis or prognosis of a specific disease, however, validation of the usefulness of such diagnostic panels still need to be done.”

Because controlled diet in the study is different from the subject’s habitual diet before the study,   miR changes in the control diet are inevitable. Furthermore our control diet is not an unhealthy diet as we described above. Even though the control group seems to have less change in miRs than the K-diet group, we cannot explain every reason of their up or downregulation due to the characteristics of miR.

Specific comments:

Critique 3) line 4  please include this was a pilot study in the title.

Ans 3) as suggested it was added to the title.

Critique 4) line 27  ten subjects seem much too low powered to detect signficance. Please consider adding n=5 to each group and re-analyzing the data.

Ans 4) We already responded to the Critique 1.

This is a pilot intervention study to determine whether K-diet can change circulating miRs in the plasma and saliva.  If we enroll 5 more subjects now to increase the statistical power, we cannot provide the exactly same diet and environment. Since we already got the fact that circulating miRs can be changed by K-diet, further validation can be done through a new study with more subjects and more specific focus on miRs related to diabetes and diabetes associated metabolism. Since we already demonstrated that salivary and plasma miRs can be changed by diet, anyone who is studying on diet can adopt this idea to their studies.

Critique 5) line 36  were any of the down-regulated miR in control diet associated with disease risk? Please include in abstract as was done with K-diet.

Ans 5) It was added to the Ln 36-37 regarding the control diet associated health condition

“which are associated with diabetes mellitus, adipogenesis and obesity”

Critique 6) line 80  please include a hypothesis based on previous literature.

Ans 6) Hypothesis was added and the whole sentence was re-written (Line 83-87).

“We, therefore, set up a hypothesis that the health effects of K-diet are conveyed through miRs, which can be detected in the circulation. To test the hypothesis we analyzed the expression changes in circulating miRs using plasma and saliva samples collected from a two-week dietary intervention study [20] and determined the metabolic influence of the K-diet, through which K-diet may promote health conditions. ”

Critique 7) line 89  is this registered as a clinical trial?

Ans 7) Unfortunately no. This small trial was done in 2016.

Critique 8) line 97  please correct, "bowl"

Ans 8) corrected in the line 105

Critique 9) line 103  please include a table listing the ingredients of each diet compared to one another.

Ans 9) Traditional drinks provided to K-diet group were green tea, burdock tea, schisandra tea, persimmon leaf tea, mugwort tea, and bokbunja tea, while the control diet group was provided with coffee, orange juice, canned soda, adlay tea, , citron tea and ginger tea,.

It seems that the reviewer suggests listing their info into a table. However, it could be very difficult to add these items precisely.

Critique 10) line 159  the sample size is small and control group started 20 mg/dL lower than k-diet which may have impacted the results. Please explain.

Ans 10) Even though the difference looks big, it is not statistically significant and the standard deviations of K-diet group is wide. The table 3 showed a lower intake of animal fat and cholesterol in the K-diet group, which is described at the Line 330-332. 

Numerically speaking, in the control group cholesterol increased by 20 mg/dL after the intervention, which is quite opposite response compared to the K-diet group that showed 40 mg/dL decrease with statistical significance. Even if the control group had the numerically same level of average cholesterol at the baseline similar to K-diet group, it may have still increased due to the higher fat component in the control diet compared to that in the K-diet..up of wat percentage change in

Critique 11) line 185  why was this miR down-regulated in control diet for both salivary and blood? Seems contradictory. Needs explaining.

Ans 11) Line 185 does not have any miR information.

Because the ‘miR down-regulated in control for salivary and blood’ is hsa-miR-25-3p, if this question is regarding hsa-miR-25-3, we already responded at the Critique 2. Please refer to Ans 2.

If the question is why this miR was down-regulated in both salivary and plasma, the answer could be found at the line 256-260.  

“Salivary glands are enveloped by capillaries and are highly permeable, allowing molecules to move into the saliva-producing cells and potentially influence the molecular constituents of the saliva [41] Furthermore, saliva contains many different components including enzymes, antibodies, hormones, cytokines and microbiota that may provide additional information regarding health and disease status.”

Critique 12) line 192  both of these seem compelling for the control to lower risk for diabetes and obesity

Ans 12) We assume that line 192 means table 6 or figure 1 because line 192 is empty. If it is regarding the down-regulation of miR -25, we have responded to the Critique 2. Please refer to the Ans 2.

Critique 13) line 206  This is not necessarily an accurate statement. Previous research suggests the possibility for food to alter dna sequence over time: Emily A. Seward, Steven Kelly. Dietary nitrogen alters codon bias and genome composition in parasitic microorganisms. Genome Biology,

2016; 17 (1) DOI: 10.1186/s13059-016-1087-9

Ans 13) Thank you for the suggestion and reference. “because nutrients/food cannot directly alter DNA sequence” has been removed.

Critique 14) line 214  do the authors have data on exosomes in the current study?

Ans 14) Unfortunately, we do not have the data.

Critique 15) line 225  and control diet for that matter.

Ans 15) The suggested information was added to the line 233.

The relationship with control diet will also add another layer of information.

Critique 16) line 228  authors need to explain why it was not validated. seems critical to the basis of this entire paragraph.

Ans 16)  

We rearranged the whole paragraph (Line 234-243) with new sentences.

We paid a special attention to this miR because it is the only miR that exhibited a polar response among the two diet groups; however, one concern is that this miR did not pass the validation. Because we cannot determine why they are different in the array but not in the RT-qPCR, we need further study.    

Critique 17) line 244  authors need to explain why these were down-regulated in the control diet. seems very discordant with their findings.

Ans 17)  We already answered to the similar critiques. We would like to compare the traditional diet and a control diet that is currently consumed and a partially Westernized Korean diet.

Since the control diet has less carbohydrate and more fat, it is not surprising that diabetes related miR can be downregulated in the control group. It suggested that our control diet influenced on a certain segment of carbohydrate metabolism. However, please refer to table 2 which demonstrated that both diet groups showed no changes in fasting glucose, insulin and HOMA-IR, it is hard to explain it in detail. Expressional level of one miR alone does not seem to be sufficient to set up diagnosis or assess disease progression.

Critique 18) line 246  authors support the need to increase the sample size prior to publication of these findings.

Ans 18) If we can increase the subject number, it could be great. Because it is not technically possible as we already mentioned above, it seems that we need future studies with more recruiting subjects based on the current data.

Critique 19) line 263  how would this be direct evidence for the mechanism underlying the metabolic effects of the k diet if the down-regulation occurred in the control diet? the authors present a confusing statement as the seeming favorable result occurred in the control diet.

Ans 19)  The whole paragraph was been changed (Line 271-279)

In the present sentence what we want to say is, if a miR change can occur in both plasma and saliva, it may give stronger evidence rather than a change only in plasma or saliva. Unfortunately a miR showed same changes in plasma and saliva by screening but was not validated.  

To avoid the confusion we removed “down-regulated in both plasma and saliva in the control group by screening.

Critique 20) line 283  this entire paragraph seems overly speculative and not directly related to the objectives of this study and should be deleted.

Ans 20) MiR can affect many biological phenomena and are involved in the development of many diseases. In the present study we found some miR changes and explained them focused on diabetes mellitus and related metabolism. However, who knows? Observed changes might be more important to the other processes such as cancer development or cardiovascular diseases. When we searched our miRs, the miRs that we found were associated not only with diabetes but also with cancer. We believe it is in the best of scientific communities’ interest to mention this possibility to the audience who has different ideas and interest.

Critique 21) line 291  this paragraph is also unnecessary and should be deleted.

Ans 21) MiR can affect many biological phenomena and are involved in the development of many diseases. In the present study, we found some miR changes and explained them focused on diabetes mellitus and related metabolism. However, who knows? Observed changes might be more important to the other processes such as cancer development or cardiovascular diseases. When we searched our miRs, the miRs that we found were associated not only with diabetes but also with cancer. We believe it is in the best of scientific communities’ interest to mention this possibility to the audience who has different ideas and interest.

Critique 22) line 299  This argument is not supported by their data. Indeed, the greater GL in the k diet would adversely affect insulin/glucose more than GI in most circumstances. The authors need a more thoughtful and appropriate explanation.

Ans 22) Even though we chose low GI diet, the GL was still higher because of higher total intake (Table 3). Nevertheless the greater GL did not significantly affect the glucose level, insulin level and HOMA-IR (Please refer to the table 2) probably due to the high consumption of fiber that may reduce carbohydrate absorption. The observed miR changes may have contributed, too (Line 309-318).

Round 2

Reviewer 1 Report

I recommend consistency with the decimals throughout the manuscript

Reviewer 3 Report

The authors have provided a thorough follow up to the suggested comments by the reviewers.

The only minor remaining comment is:

Line 35-36 needs grammatical correction.